# Family Perspectives on Newborn Screening for X-Linked Adrenoleukodystrophy in California

**DOI:** 10.3390/ijns5040042

**Published:** 2019-11-13

**Authors:** Katharina Schwan, Janey Youngblom, Kara Weisiger, Jessica Kianmahd, Rebecca Waggoner, Joanna Fanos

**Affiliations:** 1Department of Biological Sciences, California State University, Stanislaus, Turlock, CA 95382, USA; jyoungblom1@csustan.edu; 2Department of Genetics, Kaiser Permanente Oakland Medical Center, Oakland, CA 94610, USA; kara.x.weisiger@kp.org; 3Department of Pediatrics, University of California, Los Angeles, CA 90095, USA; jkianmahd@mednet.ucla.edu; 4Department of Psychology, University of California, Los Angeles, CA 90095, USA; bmwaggoner1@gmail.com; 5Institute on Disability, University of New Hampshire, Durham, NH 03824, USA; joanna.fanos@sjsu.edu

**Keywords:** X-linked adrenoleukodystrophy, ALD, newborn screening, NBS, California, family

## Abstract

X-linked adrenoleukodystrophy (ALD) is caused by gene variants in the *ABCD1* gene, resulting in a varied clinical spectrum. Males with ALD present with symptoms ranging from isolated adrenal insufficiency and slowly progressive myelopathy to severe cerebral demyelination. Females who are heterozygous for ALD typically develop milder symptoms by late adulthood. Treatment for adrenal insufficiency associated with ALD exists in the form of cortisol, and cerebral ALD may be treated with stem cell transplantation. Currently, there is no treatment for myelopathy. Since 2013, at least 14 states have added ALD to their newborn screening (NBS) panel, including California in 2016. We examined the impact of a positive NBS result for ALD on families in California. Qualitative interviews were conducted with mothers of 10 children who were identified via NBS for ALD. Interviews were transcribed verbatim and analyzed using thematic analysis by two coders. Mothers felt strongly that ALD should be included on California’s NBS panel; however, many expressed concerns over their experience. Themes included stress at initial phone call, difficulty living with uncertainty, concerns regarding mental health support, and desire for more information on disease progression, treatments and clinical trials. Mothers exhibited diverse coping strategies, including relying on faith, information seeking, and maintaining hope. Mothers’ recommendations for healthcare providers included: educating providers making the initial phone call, providing patient-friendly resources, offering information about ongoing research, and streamlining care coordination. Advice for parents of children with ALD focused on staying hopeful and appreciating the time they have with their children. As more states add ALD to their NBS panel, it is important to improve the current model to promote family resiliency and autonomy.

## 1. Introduction

X-linked adrenoleukodystrophy (ALD) is a metabolic disorder caused by a pathogenic variant in the *ABCD1* gene that results in a broad phenotype, including adrenal insufficiency and cerebral demyelination, leading to neurological deficits and progressive paralysis [1]. Patients with ALD are asymptomatic at birth; however, approximately 80% of males with ALD will develop adrenal insufficiency during their life, usually before adulthood. It is estimated that all males and many females with ALD will develop progressive myelopathy. This may present as spastic gait, spastic paraparesis, and sensory ataxia [2]. The most severe form of-ALD is cerebral ALD (cALD), which affects approximately 38% of males with the *ABCD1* gene variant [3]. Symptoms usually appear between 5–10 years of age followed by death or severe disability within several years. When cerebral manifestations appear in childhood, the earliest symptoms are typically related to cognitive dysfunction such as declining school performance and behavioral issues. Other symptoms may follow, including visual and sensory agnosia, decline in motor skills, and epileptic seizures [2]. It is estimated that more than 80% of females with the *ABCD1* gene variant may develop symptoms of myelopathy or neuropathy by age 60. Females are usually not impacted by adrenal insufficiency or cALD [3]. 

Hematopoietic stem cell transplantation (HSCT) is a well-established treatment proven to halt cerebral demyelination for boys with early-stage cALD [1]. Periodic brain MRI’s can determine whether a boy at risk for cALD is an appropriate candidate for HSCT by utilizing an MRI severity scale known as the “Loes scale” (range, 0 to 34) [4]. A score of ≥1 is considered abnormal and treatment success is highest in patients with scores ≤9 [4,5]. Transplant-related mortality is less than 5%; however, there are significant morbidity concerns including graft rejection, graft-vs-host disease, and associated long-term immunosuppression [3,6].

Given the potential benefits of early diagnosis in boys at risk for ALD, New York was the first state to add ALD to their newborn screening (NBS) panel in 2013, followed by Connecticut, California, Minnesota, and Pennsylvania. At least 14 states now screen for ALD, and many others are mobilizing to begin screening in the near future [7].

To date, evaluations of NBS programs have focused primarily on the statistical outcomes of the screening program but have failed to include longer-term clinical outcomes of children identified through ALD NBS or on the impact that positive screening results have on families. Thus, this study explores family perspectives on California’s ALD NBS program to identify areas of concern and to offer health care providers (HCPs) recommendations that go beyond treating only the medical expressions of this condition.

## 2. Materials and Methods 

Eligible participants were English-speaking adult (≥18 years) parents of children who received a positive screening result between September 2016 and January 2019 for ALD in California. Participants were identified by genetic counselors and nurse coordinators at five certified metabolic clinics: Sutter Medical Center, Sacramento; University of California, Davis; University of California, Los Angeles; University of California, San Francisco; and Valley Children’s Hospital. Interested participants reached out to the principal investigator (PI) directly for enrollment in the study. Eleven individuals were enrolled in the study, one as a pilot interview only. Consent was obtained over the phone and recorded. All participants received a $25 visa gift card. The research study was approved by the Institutional Review Board of California State University, Stanislaus (protocol #1819-004).

Interviews were conducted using an open-ended interview guide that focused on (1) understanding of the NBS result and ALD diagnosis; (2) experience receiving the NBS result; (3) family relationships; (4) coping; (5) clinical care issues; and (6) recommendations. Phone interviews were audio-recorded and transcribed by a professional transcription service. All identifying information from participants was changed to ensure confidentiality. Names of probands in the transcripts were changed to pseudonyms. The transcribed interviews were analyzed using thematic analysis by two independent coders to identify overarching themes within the data [8]. In the initial round of coding, the first two interviews were analyzed separately to classify major recurrent concepts. A preliminary codebook was established based on mutually agreed upon findings. In a second round of coding, the remaining eight transcripts were analyzed independently by both coders. These data were also compared for inter-coder reliability. Discrepancies in codes were noted and adjusted based on coder agreement.

## 3. Results

Ten mothers were interviewed about their experience with California’s NBS program for ALD. No fathers volunteered to be interviewed for this study. Demographic information is summarized in Table 1. The majority of mothers had sons (6/10) originally identified on NBS, as compared to daughters (4/10). Notably, five children were found to have a variant of uncertain significance (VUS); four were found to have a known pathogenic variant; and one had no variant identified but presented with elevated very long chain fatty acids (VLCFAs) only. As per the parent’s reports, none of the children exhibited symptoms of ALD at this time.

### 3.1. Communication of Positive NBS Results

All mothers received an initial phone call regarding the NBS screening result. Six received the results from a non-genetics provider, usually a pediatrician and in one case a nurse, while three families received the news from a geneticist. One family was unsure of the role of the individual who conveyed the initial results over the phone. Regarding prior knowledge of ALD, only one family was aware of the condition due to a previously diagnosed family memberand thus prepared for possible results found on NBS. The remaining nine families had no prior knowledge of ALD. Seven mothers reported a lack of understanding or familiarity with ALD by the HCP conveying the results. 

“I was shocked. I was in tears. She [pediatrician] didn’t seem like she knew anything. So it wasn’t like I could ask her any questions.”

Given the limited information some HCPs provided over the phone, many mothers left the initial phone call with fear and confusion regarding if and how their child was affected.

“I was a mess. I had no idea what was going on. All I knew was my daughter has this thing and she’s gonna die.”

Eight mothers responded negatively to hearing this news, describing feeling “shock”, “fear”, “anxiety”, and “confusion”. The majority of mothers (80%) reported searching the internet for ALD after learning the results. This search yielded overwhelmingly disturbing results, which often catapulted families into further distress. 

Typically, once a family was notified of the NBS result, they were referred to a specialist, usually a geneticist, for confirmatory testing. The amount of time between receiving the phone call and seeing a specialist varied significantly. Four families were seen by a specialist within 1–2 days, two within two weeks, and four families had to wait a month or more before seeing a geneticist. Typically, males identified on NBS were seen more quickly than females, unless females had older brothers. In one circumstance, a family moved homes during this time period, which may have led to the long wait time. Other factors that may have led to longer than expected wait times were limited genetics providers in the area and insurance issues. Regardless of time spent waiting, mothers reported feeling “panicked”, “stressed”, and “concerned”. 

Unlike the initial phone call, the majority of mothers (70%) reported that the information provided by the geneticist was “sufficient”. The three mothers who reported that the information provided was “not sufficient” expressed frustration that the geneticist had limited first-hand experience with ALD and thus was unable to offer more nuanced information. 

“Essentially he and the counselor just gave us a print-out of adrenoleukodystrophy. The specialist didn’t know what adrenoleukodystrophy was before Jeremy. We were the first.”

Three mothers reported feeling most reassured after meeting with the neurologist involved in the ongoing care of patients with ALD. Five mothers expressed a desire for the initial in-person consultation to include the geneticist and genetic counselor, as well as the neurologist and endocrinologist. 

“If my kids are going to be seeing neurology and endocrinology for the rest of their lives, I want one of those team members in that room with genetics. You shouldn’t have let me walk out with so much fear.”

### 3.2. Emotional Progression and Coping with the NBS Result

Interviews took place between 4 and 26 months after receiving the initial phone call. By this time, 60% of mothers reported feeling moderately or very hopeful about their child’s future, and 70% reported a positive progression in their emotions about their child’s diagnosis, even though much sadness and uncertainty remained. Differences in emotional progression were noted in mothers of sons versus daughters, with the former generally reporting more hopefulness and acceptance of their son’s result and possible ALD diagnosis when compared to mothers of daughters. This discrepancy may be explained by mothers’ misunderstanding of their daughter’s NBS result. Two of the three mothers with daughters expressed concern that their child may become severely disabled or die in childhood, even though females generally develop only the milder symptoms of myelopathy in adulthood.

Mothers who experienced a positive emotional progression regarding their child’s NBS result utilized varied coping approaches, including relying on faith, seeking information, and maintaining hope. Half of the mothers mentioned the importance of religion in their ability to adjust to their child’s diagnosis. 

“Religiously, I don’t believe that just because you die it’s the end of that life, so it’s not as much of a loss.”

Information seeking was reported by 7/10 mothers as a major coping strategy. This included going back to school, actively engaging with the ALD community, attending conferences, researching and participating in clinical trials, and gaining an in-depth understanding of the condition, possible progression, and outcomes. 

Mothers of both boys and girls struggled with a lack of concrete answers regarding their child’s NBS result and possible disease progression.

“I think that’s the worst part, because you don’t know. They told me they expect her to have a normal childhood, but then what if she doesn’t? And if she goes all her life and she has normal childhood and then what if it hits in her adult years and she can’t live the way that she wants to live. As a parent, you already have a lifetime responsibility. But, now, I’ve created a long-term responsibility for myself.”

One mother reported denial of potential future health impacts on affected sons to better manage the painful emotions associated with this train of thought.

“I try not to think about the future too much because it freaks me out. I try to envision positive things, but if I think too far in the future sometimes my mind goes: What if he’s in a wheelchair? What if he’s dead? So I really don’t think about it very often. I guess that’s the way that I deal with it.” 

At the time of these interviews, none of the children identified on NBS were symptomatic. Thus, although uncertainty was often mentioned as a major source of stress, it also offered mothers hope: hope that their child may never develop the fatal form of ALD, as well as hope that, given their young age, better treatments and potential cures would become available.

“In 10, 20 years there may be a medication. Or the gene therapy is going to be the gold standard and we’re good. I do feel lucky that he is so young and we have time on our side.”

### 3.3. Impact on Family Dynamics

Four mothers reported that having the ALD diagnosis in the family negatively impacted the relationship with their spouse, three reported a positive impact, and three reported no impact. In two of the families reporting a negative impact, the father was found to carry a pathogenic *ABCD1* variant and was diagnosed with ALD. As a result, two mothers found themselves concerned about both their daughter’s and their partner’s health. One mother shared the difficulty her husband was having with his own diagnosis and how it affected their relationship. 

“He [husband] went into a depression. We didn’t have any means of communication. I eventually got him to go to therapy, but even during his time in therapy, he wasn’t coming home and talking to me. It’s been a rough almost two years.”

Blaming one another for the diagnosis in the family, the burden of juggling countless medical visits, denial of the diagnosis on behalf of male partners, and deciding to stop trying for more children were also reported as factors that negatively influenced the relationship. One mother pointed out that inequality between her and her partner’s role as health advocates for their child created distance in their marriage. 

“I noticed almost immediately that it’s all about the mom. I became my son’s health advocate, a role that I never really anticipated playing. Most of the stress falls on me. I schedule all his appointments. I have all the conversations. I’m a member of the support groups. And he [husband] doesn’t. So that’s a wedge.”

Of the mothers who reported a positive impact, all mentioned that receiving the diagnosis created stronger bonds and appreciation for one another. Mothers who reported a team-based approach to parenting also felt more positive about their relationship.

Most mothers reported that learning the NBS result impacted their parenting style. The eight mothers who adapted their parenting described their style as more protective, loving, indulgent, and attentive. Seemingly minor health concerns were a greater source of anxiety for these mothers, given the potential association with ALD. 

“When Justin would get a rash, I would be like, ‘Oh it will go away’, but when Forrest has a fever or anything, it’s really alarming.”

Although mothers did not express favoritism among their children, many did acknowledge that potentially affected children received more attention and were disciplined less harshly than older siblings had been.

### 3.4. Overall Opinion on the NBS Process and Follow-Up Care

Overwhelmingly, all participants were in favor of including ALD on the NBS panel for both males and females; however, the timing of when testing should be offered varied. Sixty percent of mothers believed that testing for ALD in males should be offered prenatally, and 40% believed it should be offered at birth. For females, 30% believed that testing should be offered prenatally and 70% believed it should be offered at birth. 

All affected males were followed by endocrinology and neurology in 6 to 12-month intervals. Of nine families in whom a gene variant was identified in a male, either a son or a father, seven mothers were satisfied with this care plan. Mothers of daughters conceptually understood that their children were not at risk for developing the cerebral form of ALD. However, concern about lack of follow-up care and possible symptoms in childhood were mentioned by two mothers, suggesting additional support and reassurance may be necessary.

“I’m just curious if Clarissa does need to be seen more often. I feel like because she’s female and she’s not presenting right now, they just pushed her to the back burner. I’m left over here wondering, what if she is presenting and I don’t know?”

### 3.5. Recommendations for Healthcare Providers and Families

The most common advice mothers had was for pediatricians to educate themselves prior to calling families with the NBS results. Alternatively, mothers believed the initial phone call should be made by a specialist with more knowledge of the condition. Three mothers mentioned the need for more mental health resources. Having genetic counselors present for the meeting with the geneticist was appreciated by three mothers. However, an additional two mothers would have preferred to see other specialists at the initial visit as well, in particular the neurologist. Mothers of daughters desired further guidance on how to speak to their children about ALD, in particular on broaching the topic of family planning. Mothers desired more patient-friendly resources on ALD, including information on disease progression, clinical trials, care coordination, and contact lists for specialists with ALD experience. Three mothers asked for information on financial resources for cascade testing and other services not covered by insurance. Lastly, five mothers felt that the time from initial NBS result to official diagnosis was too long. This could potentially be shortened by including *ABCD1* sequencing on the NBS. Table 2 summarizes mothers’ recommendations for the HCPs.

Normalization was common advice that mothers offered to other families. They felt it was important to recognize that ALD is a disease like any other, and that parents should appreciate the time they have with their children. One mother expressed concern that learning this diagnosis so early in a child’s life might impact bonding with their newborn and encouraged parents to develop loving relationships with their affected children. Lastly, mothers recommended avoiding the internet to learn more about ALD.

## 4. Discussion

This study addressed a significant gap in the literature by elucidating the unique psychosocial and clinical experiences of parents with children who screened positive for ALD on NBS. 

### 4.1. Educating Healthcare Providers on ALD

High stress and anxiety at initial phone call were reported by nearly all participants in this study. These results are consistent with studies looking at the parental NBS experience for other diseases, such as Pompe disease (PD) and cystic fibrosis [9,10]. Given the myriad of negative emotions experienced by parents upon learning their child’s positive screening results, HCPs who work with these families should be aware of the factors that contribute to a positive, informative, and supportive NBS experience, in the context of a potentially devastating diagnosis. This sentiment is shared by parents of children who received positive NBS results for PD, as studied by Pruniski et al. [9]. As with ALD, the parents of these children felt that the HCP making the initial disclosure was not fully informed on PD. These families also emphasized education on the disease as an important recommendation for both HCPs and parents [9]. For ALD, emphasizing currently available treatment and possible future treatment may be beneficial, as well as pointing out that newborns identified on NBS are not at any immediate risk for health issues. Parents may find comfort in knowing that although future risks are serious, at the time of NBS they need not worry about any sudden changes in their child’s health. For females, parents should understand that their daughter is not at risk for cALD or adrenal insufficiency and that any future symptoms of myelopathy would be on the milder side and not until adulthood.

In a study exploring physician’s experience notifying parents of positive cystic fibrosis NBS results, physicians reported that the NBS program could provide more resources to aid them in “breaking bad news” to patients, including training on having these difficult conversations, an information sheet to distribute to families, as well as a FAQ sheet with sample responses [11]. These types of interventions would likely benefit pediatricians revealing ALD results as well and could aid in streamlining the initial disclosure process. 

### 4.2. Supporting Healthy Coping

Szulczewski et al. found that caregivers of children with pediatric chronic diseases showed decreased psychological functioning when uncertainty was high [12]. Similarly, the authors found that less uncertainty was associated with improved coping mechanisms. Therefore, reducing anxiety as much as possible should be an integral aspect of counseling around ALD; for example, by assuring families that NBS is designed to catch diseases early when treatment is most effective.

HCPs are uniquely positioned to support parents in continuing and adopting healthy coping mechanisms, particularly around seeking information. By providing patient-friendly resources, parents may feel more confident in their ability to care for their child. By providing information on support groups and clinical trials, parents may increase their social network. Reliance on faith was one of the strongest predictors of healthy coping. Most hospitals provide chaplains who can engage with patients’ spiritual and religious beliefs [13]. 

### 4.3. Diagnosing Fathers

Adult males learning about their diagnosis of ALD as a result of a daughter’s positive result was an unexpected outcome of this screening program. Although nearly all men will experience some form of adrenal issues and myelopathy in their lifetime, some may be spared the more disabling symptoms of neurological dysfunction. However, approximately 20% of affected males may develop more severe symptoms after age 18 that could include progressive adrenomyeloneuropathy with cerebral manifestations [14]. As represented by one of the fathers in this study, an ALD diagnosis in an adult male could lead to depression. Despite multiple medical visits, mental health support is not part of the follow-up process for affected males or other family members. Thus, offering mental health resources to newly diagnosed adult males, as well as any other family member impacted by ALD, should be an integral aspect of follow-up care post NBS result.

### 4.4. Limitations

This study was limited to ten mothers of children identified on NBS for ALD in the state of California. Although fathers were eligible to participate in the study, only mothers volunteered. Therefore, the paternal perspective is not included in this study. Participants were primarily Caucasian and English-speaking, and most reported some form of college education. Recruitment for this study was limited to families who were seen by one of the five metabolic centers that participated in this study. In addition, classification of the gene variant and diagnosis as known pathogenic, VUS, or elevated VLCFA stemmed from the parent report and was not confirmed with medical records. None of the parents of children with a false-positive result or parents of children diagnosed with an untreatable peroxisomal disorder participated in this study, although these individuals may also be identified through ALD NBS. Additionally, given that NBS for ALD was only added to California in 2016, these families were still in the early stages of their experience with ALD with no symptomatic children identified as of yet. Thus, the results of this study may not be generalizable to families who have older children diagnosed with ALD through NBS in other states and who have started showing symptoms of the disease. Additionally, there is the possibility of participation bias playing a significant role in this research study. Presumably, given the small sample size, many families were contacted and chose not to participate.

### 4.5. Research Recommendations

Future studies that include the paternal perspective as well as the first-hand experience of fathers diagnosed with ALD as a result of their daughter’s NBS result would yield revealing information on the impact of cascade screening and the opinions of fathers. Considering the diversity of California’s population, it is important that future studies on this topic explore non-English speakers’ experience with NBS, as well as those of parents with more diverse backgrounds in terms of socioeconomic status, education, race, and ethnicity. A follow-up study interviewing families whose children are further along in their disease course may uncover additional psychosocial and medical concerns not elucidated in this early study. Lastly, this study is unique to California’s NBS screening process, protocols, and patient experience, and future studies looking at the NBS experience in other states will likely yield different informative data.

## 5. Conclusions

This study contributes to the limited literature that exists about NBS for ALD, and its findings may be used by HCPs and policymakers to tailor the support, counseling, and clinical needs of these unique patients and families. Our findings suggest that the initial phone call revealing the positive results to parents had a strong influence on parental emotional reactions and understanding of the result. Thus, educating HCPs on ALD prior to revealing the NBS result was the most common recommendation. Ongoing visits with specialists, particularly neurologists, provided some clarity on the diagnosis. A greater emphasis on providing mental health support to all members of the family needs to be considered as an essential part of follow-up care for ALD, particularly as cascade screening is expanded to parents, siblings, and distant relatives for whom these results could have serious clinical and emotional consequences. 

As more states add ALD to their NBS panel, it is increasingly important to improve the current model to emphasize providing detailed condition and treatment information as well as comprehensive resources that promote family resilience and autonomy.

## Figures and Tables

**Table 1 IJNS-05-00042-t001:** Demographic information.

Participant Demographics	Response	*n*
Parent Interviewed	Mother	10
Ethnicity	Caucasian	5
	African American	1
	Latina	2
	Mixed	2
Marital Status	Married	9
	Partnered	1
Education	Graduate Degree	1
	College Degree	4
	Associate Degree	3
	Some College, No Degree	1
	High School Degree or GED	1
Proband’s Gender	Male	6
	Female	4
Family Members of Proband	Proband has siblings	10
Newborn Screening Result	Pathogenic variant	4
	Variant of Uve	5
	Elevated VLCFA only	1

**Table 2 IJNS-05-00042-t002:** Recommendations for Healthcare Providers.

Education of HCPs, particularly pediatricians, on ALD Shorter wait times between initial phone call and meeting with specialist Faster turnaround times for the confirmatory genetic testing result, possibly by including *ABCD1* sequencing on NBS Meeting with specialists to include geneticist, genetic counselor, neurologist, and endocrinologist Resources and support in coordinating follow-up care Referrals and/or resources for mental health services Additional resources on ALD, including printed materials, online materials, and support groups Information on clinical trials Contact list for specialists with experience in ALD Guidance/resources on discussing ALD diagnosis with children Financial resources for cascade testing and other medical care if limited insurance coverage

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
