# Peer review of "Family Perspectives on Newborn Screening for X-Linked Adrenoleukodystrophy in California"

_2409-515X, 2019, doi:10.3390/ijns5040042_

Round 1

Reviewer 1 Report

This manuscript is a nice read with an original and important message. However, an insufficient amount of data and/or questionnaires have been presented to make it of real scientific importance. The words "several" and "some" are used often. Please specify the exact numbers/percentages. There is no referral to collected data.

Furthermore, some references are quoted incorrectly and some medical content is incorrect. 

Line 18: "Since treatment for X-ALD exists in the form of stem cell transplantation, ten states added X-ALD to their newborn screening (NBS) panel, including California". This is a strange sentence and it has errors. The rational for NBS is the availability of a treatment for both cerebral ALD (hematopoietic stem cell transplantation), but also adrenal insufficiency (cortisol). Myelopathy can not be treated (yet). Currently 14 states are screening. 

Line 42: Heterozygous females are usually not impacted by cALD (n)or adrenal insufficiency

Line 46: periodic cerebral MRI's are used to perform LOES-scores. The LOES score is used to determine whether a boy is eligible for a HSCT or not.

Line 49: Incorrect quote. The referenced source states: "However, early diagnosis of boys with adrenoleukodystrophy can lead to life-saving interventions. These include initiating timely adrenal steroid replacement therapy following detection of adrenal insufficiency, and for providing allogeneic hematopoietic stem cell transplantation (HSCT) as a means of treating cerebral ALD. HSCT can arrest the often fatal progression of cerebral demyelination provided that the procedure is performed at a very early stage of the disease. Unfortunately, this can only be effective during a narrow therapeutic window, which is often missed. Newborn screening provides access to this “window of opportunity” and allows for timely initiation of these established therapies."

and

"In April 2012, following the death of their son, Aidan, who had cerebral ALD, but was diagnosed too late, the Seeger family drafted and supported the passage of Aidan’s Law in the State of New York. The bill was approved in February 2013 and became law in March 2013. On 30 December 2013, New York State’s newborn screening laboratory began testing babies for adrenoleukodystrophy."

Line 53: "To date, there are no evaluations of NBS programs for X-ALD." Be more specific on which aspects, since there is elaborate literature on screening algoritms/results from several states, like New York State and Minnesota. 

Results: 

- Would you expect different outcome if you interviewed the fathers of the same children? What was the rational for only interviewing the mothers?

Line 82: Of the 10 families, 5 had a VUS (were functional studies in skin fibroblasts etc performed? Or are these cases unresolved?), 4 had a pathogenic variant (=ALD) and in 1 no ABCD1 variant was identified, but showed elevated VLCFA. In the absence of a variant in ABCD1, the newborn may have another disease associated with elevated VLCFA (ACOX1, DBP, Zellweger, AGS, etc.). Is there any information on follow up? This newborn possibly does not have ALD. 

- A lot of children's names are used. Please anonymize!! (my boy, my child) or pseudonymize and mention that you did this. 

- Line 119: Interviews took place between 4 and 26 months. Does this range have influence on the results? Since the range is very wide which might mean variation in the number of consultations etc will be big? This may very well affect the perspective.

- Line 138: Here the authors speak of parents, but everywhere else throughout the manuscript only "mothers" is used. Were the fathers included in this question? Should it be: mothers stated it offered them and their husbands etc. 

- Line 139: the quote is from one person. Is this answer also given by other mothers? Since your result states parents. And is there a difference between parents of boys and girls? Since girls usually only develop a myelopathy.      - Line 153-157 Is this problem in the same family, in both families?                       

Author Response

This manuscript is a nice read with an original and important message. However, an insufficient amount of data and/or questionnaires have been presented to make it of real scientific importance. The words "several" and "some" are used often. Please specify the exact numbers/percentages. There is no referral to collected data.

Furthermore, some references are quoted incorrectly and some medical content is incorrect. 

Line 18: "Since treatment for X-ALD exists in the form of stem cell transplantation, ten states added X-ALD to their newborn screening (NBS) panel, including California". This is a strange sentence and it has errors. The rational for NBS is the availability of a treatment for both cerebralALD (hematopoietic stem cell transplantation), but also adrenal insufficiency (cortisol). Myelopathy can not be treated (yet). Currently 14 states are screening. 

Updated per reviewer suggestion

Line 42: Heterozygous females are usually not impacted by cALD (n)or adrenal insufficiency

Updated per reviewer suggestion

Line 46: periodic cerebral MRI's are used to perform LOES-scores. The LOES score is used to determine whether a boy is eligible for a HSCT or not.

Updated per reviewer suggestion

Line 49: Incorrect quote. The referenced source states: "However, early diagnosis of boys with adrenoleukodystrophy can lead to life-saving interventions. These include initiating timely adrenal steroid replacement therapy following detection of adrenal insufficiency, and for providing allogeneic hematopoietic stem cell transplantation (HSCT) as a means of treating cerebral ALD. HSCT can arrest the often fatal progression of cerebral demyelination provided that the procedure is performed at a very early stage of the disease. Unfortunately, this can only be effective during a narrow therapeutic window, which is often missed. Newborn screening provides access to this “window of opportunity” and allows for timely initiation of these established therapies."

and

"In April 2012, following the death of their son, Aidan, who had cerebral ALD, but was diagnosed too late, the Seeger family drafted and supported the passage of Aidan’s Law in the State of New York. The bill was approved in February 2013 and became law in March 2013. On 30 December 2013, New York

State’s newborn screening laboratory began testing babies for adrenoleukodystrophy."

Updated per reviewer suggestion

Line 53: "To date, there are no evaluations of NBS programs for X-ALD." Be more specific on which aspects, since there is elaborate literature on screening algoritms/results from several states, like New York State and Minnesota. 

Updated per reviewer suggestion

Results: 

- Would you expect different outcome if you interviewed the fathers of the same children? What was the rational for only interviewing the mothers?

Participants reached out to me (the PI) directly after receiving information on the study from GCs/nurses at the site they were seen. Although both mothers and fathers were eligible to participate, no fathers reached out to participate in the study. This is clarified in the methods and limitations sections.

Line 82: Of the 10 families, 5 had a VUS (were functional studies in skin fibroblasts etc performed? Or are these cases unresolved?), 4 had a pathogenic variant (=ALD) and in 1 no ABCD1 variant was identified, but showed elevated VLCFA. In the absence of a variant in ABCD1, the newborn may have another disease associated with elevated VLCFA (ACOX1, DBP, Zellweger, AGS, etc.). Is there any information on follow up? This newborn possibly does not have ALD.

This information was provided by the parent and was not confirmed with medical records. At the time of the interviews, none of the children had symptoms yet and unfortunately I do not have any follow up information. It is unlikely that the patient with elevated VLCFA has Zellweger’s because she was nearly 2 at the time of the interview and had no symptoms. One mother reported confirmation via skin fibroblasts but she specifically asked me not to include this in the data because she felt it could be identifiable information. I included a limitations section which addresses the fact that other peroxisomal disorders can be picked up via NBS but that no parents of children with other peroxisomal disorders were interviewed in this study.

A lot of children's names are used. Please anonymize!! (my boy, my child) or pseudonymize and mention that you did this. 

These are pseudonyms. I clarified this in the methods section.

Line 119: Interviews took place between 4 and 26 months. Does this range have influence on the results? Since the range is very wide which might mean variation in the number of consultations etc will be big? This may very well affect the perspective.

Interestingly, my results do not show a big difference in emotional progression based on timing. I did see a difference between mothers of boys versus girls, which is now noted in the results.

Line 138: Here the authors speak of parents, but everywhere else throughout the manuscript only "mothers" is used. Were the fathers included in this question? Should it be: mothers stated it offered them and their husbands etc. 

Made change to mothers.

Line 139: the quote is from one person. Is this answer also given by other mothers? Since your result states parents. And is there a difference between parents of boys and girls? Since girls usually only develop a myelopathy. 

This quote is an example of the concerns of a mother from a boy. However, in 2 out of 3 cases, the mothers of girls were also very concerned about their daughter’s health status because they did not truly understand what ALD means in a female. Provided a quote from a mother of a girl to elucidate this finding.

Line 153-157 Is this problem in the same family, in both families?

Fear for the partner’s health was present in both families. Depression was only found in one father because he had symptoms of myelopathy which could now be explained by his ALD diagnosis. This has been clarified.

Reviewer 2 Report

The authors summarize qualitative interviews with mothers of infants identified by newborn screening to have adrenoleukodystrophy (ALD). ALD is a heterogeneous condition with variable presentations and ages of presentation. There is little written in the literature about the parent perspective with newborn screening. As the ALD newborn screen, not only identifies the baby as being at risk for disease, but also often identifies multiple family members at risk for ALD, this initial report is valuable for drawing attention to the need for improved communication with families and care centers. The report also notes the changes in parenting style and family dynamics that can occur with a chronically sick child and supports the need for comprehensive care of child and family following the positive newborn screen.

Broad Comments:

Because there are no prior reports of parent perspective for ALD newborn screening, it would be helpful to reference other studies that have assessed parental experience with newborn screening for other conditions. Example: Rueegg CS, Barben J, Hafen GM, et al. Newborn screening for cystic fibrosis – The parent perspective. J Cyst Fibros. 2016;15(4):443-451. Background: It may be helpful to also discuss that almost all males develop neurologic disease and if not cerebral ALD, then adrenomyeloneuropathy (AMN), which is also devastating but typically starting in young adulthood. It would also be helpful to explain that adrenal insufficiency is life-threatening and requires taking multiple doses of hydrocortisone daily and training for administration of stress doses of hydrocortisone. Results: Can you comment on how many positive screens California had over the period of the study and how these 10 mothers were selected? Were there parents who were approached to participate in the study who declined participation? It seems like there could be bias involved in parents who either agreed or disagreed to participate. Can the authors comment on the potential bias? Can the authors comment on whether a diagnosis was given to the one patient with elevated VLCFA and no variant in the ABCD1 gene? Children fitting into this category often have another peroxisome disorder, such as Zellweger syndrome, which does not have treatment options available for the neurologic disease. Including this parent in the study may yield a much different perspective. California and New York have the ABCD1 sequencing as part of the blood spot screen (tier 3), which helps move the diagnostic process along at a faster pace. It may be worth noting that ABCD1 sequencing is not available on all of the newborn screens (Connecticut and Minnesota do not offer gene sequencing from the blood spot) and that sequencing as to be performed at outside laboratories (typically takes a few weeks). It’s notable that even in a state with genetic test results available the parents reported diagnostic delay as a factor causing anxiety. Can the authors comment on the time from initial notification to time of visit with the geneticist to confirm diagnosis? It may also be worth noting that including the ABCD1 sequencing on the newborn screen can reduce time to diagnosis. While it would be ideal to have geneticists, genetic counselors, neurologists and endocrinologists all see the patient at the initial visit, because ALD newborn screening could potentially identify conditions other than ALD, having neurology and endocrinology present for initial visits does not seem practical. The Pediatric Endocrine Society publication, Regelmann MO, Kamboj MK, Meller BS, et al. Adrenoleukodystrophy: Guidance for Adrenal Surveillance in Males Identified by Newborn Screen. J Clin Endocrinol Metab. 2018;103(11):4324-4331, suggests an “Urgent Referral to Pediatric Endocrinology” following diagnostic confirmation with genetics in the algorithm.

Specific Comments:

The preferred abbreviation is ALD, rather than X-ALD. While ALD is on the X chromosome, because females heterozygous for ABCD1 mutations are also affected later in life, there has been a movement to drop the qualifier of “X-linked.” Abstract/Line 51 - The expansion of newborn screening has been relatively rapid. As of October 28, 2019, there are 14 states performing newborn screening. Given the time it takes to get to publication and the speed with which ALD newborn screening is expanding, it may be best to rephrase to “At least 14 states,” rather than “Ten states” Line 41 and 43, authors should add “years” after ages. Because this is a newborn screening article, it should be clear that it typically takes years for neurologic symptoms to develop, rather than months.

Author Response

The authors summarize qualitative interviews with mothers of infants identified by newborn screening to have adrenoleukodystrophy (ALD). ALD is a heterogeneous condition with variable presentations and ages of presentation. There is little written in the literature about the parent perspective with newborn screening. As the ALD newborn screen, not only identifies the baby as being at risk for disease, but also often identifies multiple family members at risk for ALD, this initial report is valuable for drawing attention to the need for improved communication with families and care centers. The report also notes the changes in parenting style and family dynamics that can occur with a chronically sick child and supports the need for comprehensive care of child and family following the positive newborn screen.

Broad Comments:

Because there are no prior reports of parent perspective for ALD newborn screening, it would be helpful to reference other studies that have assessed parental experience with newborn screening for other conditions. Example: Rueegg CS, Barben J, Hafen GM, et al. Newborn screening for cystic fibrosis – The parent perspective. J Cyst Fibros. 2016;15(4):443-451.

Included the CF study in the results section, along with study on parental experience with Pompe disease (Pruniski et al 2018).

Background: It may be helpful to also discuss that almost all males develop neurologic disease and if not cerebral ALD, then adrenomyeloneuropathy (AMN), which is also devastating but typically starting in young adulthood. It would also be helpful to explain that adrenal insufficiency is life-threatening and requires taking multiple doses of hydrocortisone daily and training for administration of stress doses of hydrocortisone.

Provided more background on ALD symptoms in males and females and provided some information on treatment for adrenal insufficiency.

 Results: Can you comment on how many positive screens California had over the period of the study and how these 10 mothers were selected?

I do know what the number of positive screens is for this study period. I provided more detail in methods section on recruitment. I recognize that the sample size is very small and there is likely some selection bias. This is addressed in the limitations section.

Were there parents who were approached to participate in the study who declined participation?

Parents reached out to me to participate, so I only heard from interested participants. No one dropped out after they agreed to participate.  

It seems like there could be bias involved in parents who either agreed or disagreed to participate. Can the authors comment on the potential bias?

Included selection bias in the limitations paragraph.

Can the authors comment on whether a diagnosis was given to the one patient with elevated VLCFA and no variant in the ABCD1 gene? Children fitting into this category often have another peroxisome disorder, such as Zellweger syndrome, which does not have treatment options available for the neurologic disease.

At the time of the interview, the child did not have any symptoms or a diagnosis, therefore unlikely she had Zellweger’s. Unfortunately I do not have follow up information for confirmation.

Including this parent in the study may yield a much different perspective.

This parent is included in the study. Limitations sections includes the fact that children with other diagnosed peroxisomal disorders were not included in this study.

California and New York have the ABCD1 sequencing as part of the blood spot screen (tier 3), which helps move the diagnostic process along at a faster pace. It may be worth noting that ABCD1 sequencing is not available on all of the newborn screens (Connecticut and Minnesota do not offer gene sequencing from the blood spot) and that sequencing as to be performed at outside laboratories (typically takes a few weeks).

Provided more information on the CA screening program in the background section.

It’s notable that even in a state with genetic test results available the parents reported diagnostic delay as a factor causing anxiety. Can the authors comment on the time from initial notification to time of visit with the geneticist to confirm diagnosis?

Included more information on timing in the Communication on NBS section.

It may also be worth noting that including the ABCD1 sequencing on the newborn screen can reduce time to diagnosis.

Included this suggestion in recommendations.

While it would be ideal to have geneticists, genetic counselors, neurologists and endocrinologists all see the patient at the initial visit, because ALD newborn screening could potentially identify conditions other than ALD, having neurology and endocrinology present for initial visits does not seem practical. The Pediatric Endocrine Society publication, Regelmann MO, Kamboj MK, Meller BS, et al. Adrenoleukodystrophy: Guidance for Adrenal Surveillance in Males Identified by Newborn Screen. J Clin Endocrinol Metab. 2018;103(11):4324-4331, suggests an “Urgent Referral to Pediatric Endocrinology” following diagnostic confirmation with genetics in the algorithm.

Thank you for this information. I agree with your assessment but included this in recommendations to HCPs because it was voiced by several parents.

Specific Comments:

The preferred abbreviation is ALD, rather than X-ALD. While ALD is on the X chromosome, because females heterozygous for ABCD1 mutations are also affected later in life, there has been a movement to drop the qualifier of “X-linked.” This has been updated

 Abstract/Line 51 - The expansion of newborn screening has been relatively rapid. As of October 28, 2019, there are 14 states performing newborn screening. Given the time it takes to get to publication and the speed with which ALD newborn screening is expanding, it may be best to rephrase to “At least 14 states,” rather than “Ten states” This has been updated

Line 41 and 43, authors should add “years” after ages. Because this is a newborn screening article, it should be clear that it typically takes years for neurologic symptoms to develop, rather than months. This has been updated.